# Adsorption of an Anionic Surfactant (Sodium Dodecyl Sulfate) from an Aqueous Solution by Modified Cellulose with Quaternary Ammonium

**DOI:** 10.3390/polym14071473

**Published:** 2022-04-05

**Authors:** Ming Zou, Haixin Zhang, Naoto Miyamoto, Naoki Kano, Hirokazu Okawa

**Affiliations:** 1Graduate School of Science and Technology, Niigata University, 8050 Ikarashi 2-Nocho, Nishi-ku, Niigata 950-2181, Japan; zouuming330@gmail.com (M.Z.); f21k005k@mail.cc.niigata-u.ac.jp (H.Z.); 2Department of Chemistry and Chemical Engineering, Faculty of Engineering, Niigata University, 8050 Ikarashi 2-Nocho, Nishi-ku, Niigata 950-2181, Japan; nmiyamoto@eng.niigata-u.ac.jp; 3Department of Applied Chemistry, Faculty of Science and Engineering, Akita University, Tegatagakuenmachi 1-1, Akita 010-0852, Japan; okawa@mine.akita-u.ac.jp

**Keywords:** anionic surfactant, lauryl sodium sulfate (SDS), quaternary ammonium cation, modified cellulose, adsorption

## Abstract

In this study, a method of removing an anionic surfactant sodium dodecyl sulfate (SDS) from an aqueous solution by cellulose modified with quaternary ammonium cation was discussed. Cellulose, as the adsorbent, was obtained from medical cotton balls, and the quaternary ammonium cation (synthesized from dodecyl dimethyl tertiary amine and epichlorohydrin) was grafted onto the sixth hydroxyl group of D-glucose in the cellulose by the Williamson reaction under alkaline conditions. The modified cellulose was characterized by Fourier transform infrared spectroscopy (FT-IR), scanning electron microscopy (SEM), and X-ray photoelectron spectroscopy (XPS); and the zeta potential of the material was also measured after confirmation of the synthesis of quaternary ammonium salts by nuclear magnetic resonance (NMR). From these analyses, a peak of the quaternary ammonium group was observed at 1637 cm^−1^; and it was found that the surface of the material exhibited a positive charge in pH 2–7. The optimal conditions for SDS adsorption by modified cellulose were pH of 7, contact time of 3 h, and temperature of 60 °C in this study. Typical adsorption isotherms (Langmuir and Freundlich) were determined for the adsorption process, and the maximal adsorption capacity was estimated as 32.5 mg g^−1^. The results of adsorption kinetics were more consistent with the pseudo-second-order equation, indicating that the adsorption process was mainly controlled by chemical adsorption. Furthermore, thermodynamic analysis indicated that the adsorption process of SDS on the modified cellulose was endothermic and spontaneous and that an increasing temperature was conducive to adsorption.

## 1. Introduction

In recent years, with the improvement of people’s lives, the impact on the environment is also increasing. The discharge of industrial sewage and domestic sewage causes water pollution. Water pollution sickens 250 million people every year. In the period of rapid development, Japan experienced a number of sanitation incidents due to water pollution, such as Minamata disease (Hg) and Itai-itai disease (Cd). In addition, with the outbreak of the 2011 Great East Japan Earthquake, a large amount of radioactive cesium sewage produced from the Fukushima First Nuclear Power Plant was reported [1]. 

Surfactant molecules are composed of hydrophilic and hydrophobic groups, which are widely used in cosmetics, detergents, and other everyday products as emulsifiers, foaming agents, and cleaning agents [2,3]. The output value of surfactants in the world reached 47.6 billion US dollars in 2020, an increase of 8.68% compared with the previous year [4]. The yield increased from 3000 tons in 1950 to 17.8 million tons in 2015 [5,6,7,8,9]. Anionic surfactants account for the highest proportion, at about 45%. Although surfactants are important chemical products, when they are discharged into the environment, they produce a large number of bubbles, blocking the exchange of oxygen between water and air, leading to the destruction of the living environment of aquatic organisms, thus affecting biodiversity.

In the past, various adsorbents of surfactants have been widely studied. For example, Yang et al. [10] determined the most suitable method of using acrylic resin to adsorb surfactants by comparing the adsorption properties of directly purchased resin XAD-7 and an acrylic resin with different pore sizes. In addition, the adsorptive removal of sodium dodecyl sulfate (SDS) using activated coconut shell was discussed by Pravin et al. [11]. Furthermore, research into improving the adsorption capacity of surfactants by modified activated carbon has also been carried out in our laboratory. With the rapid development of new materials, metal–organic frameworks (MOFs) have received more attention in recent years. MOFs as a new porous material, also possess excellent adsorption properties. Mozhgan et al. [12] investigated the adsorption of SDS using hierarchical porous zeolitic imidazolate framework-8/carbon fiber (ZIF-8/carbon fiber) nano sorbent. MOFs are also widely used for the adsorption of other substances. For example, selective dye adsorption by UiO-66 MOF [13], adsorption of heavy metal ion Cr(VI) on UiO-66-NH_2_ MOFs [14], or chemical fixation of CO_2_ by amine functionalized MOF [15] have been reported. Modifying a material with some functional groups can be one of the main methods of achieving efficient surfactant adsorption [16]. At present, the volume of wastewater including surfactants that need treatment is excessive, and thus we still need to explore adsorption methods that achieve stronger adsorption capacity with cheaper raw materials. The methods for treating environmental pollutants through modified natural materials have been reported [17,18,19]. It can be considered that modified cellulose is also being used to remove dyes and heavy metal ions from the wastewater. However, there have been few studies on the use of cellulose-based adsorbents for the adsorption of surfactants. Thus, in this study, we paid attention to synthesizing a new type of adsorption material based on cellulose as a natural polymer material, which is expected to be environmentally friendly and inexpensive for removing anionic surfactants. Specifically, we tried to synthesize the adsorbent by modifying a quaternary ammonium cation of cellulose.

Cellulose is the main component of plant cell walls and plant fibers and is the most abundant carbohydrate on earth. The annual output of cotton, which has more than 90% cellulose content, is 26 million tons [20], and the annual recovery of wastepaper, of which the main component of cellulose, is 240 million tons [21]. Cellulose is a polymer composed of glucose monomers, and the molecules consist of hydrogen bonded into a seal form. The sixth hydroxyl group of D-glucose in cellulose has high activity and reacts easily with other substances [22]. On the other hand, quaternary ammonium cations are positively charged polyatomic ions that are generally synthesized by tertiary amines. Thus, quaternary ammonium salts with the positive charge can attract the hydrophilic anion of anionic surfactants such as SDS by electrostatic gravitation as well as by van der Waals forces to attract the hydrophobic functional group. For example, N,N-dimethyldodecylamine was selected in our experiments, expecting to enhance the adsorption efficiency through the interaction between the dodecyl group of quaternary ammonium cation and SDS.

The objective of the present research was to investigate the SDS adsorption efficiency of modified cellulose in water, aiming to develop a method for the efficient adsorption of SDS and apply it in practice. We first synthesized an environmentally friendly, low-cost, and effective adsorbent of anionic surfactants by using a natural polymer material based on cotton. In this process, it was important to select an appropriate tertiary amine that reacted with epichlorohydrin for synthesizing the corresponding quaternary ammonium cation. The quaternary ammonium cation in cellulose was modified by the Williamson reaction under alkaline conditions. Subsequently, the characteristics of the modified cellulose were explored, and the SDS adsorption mechanism of the modified cellulose was investigated in this study.

## 2. Materials and Methods 

The cellulose was derived from Osaki cotton balls from Osaki Medical Co., Ltd. (Nagoya, Japan) in this study. The sodium dodecyl sulfate (SDS) and epichlorohydrin were sourced from Tokyo Chemical Industry Co., Ltd. (Tokyo, Japan), and the dodecyl dimethyl tertiary amine was from FIJIFILM Wako Chemical Co., Ltd. (Osaka, Japan). All reagents used were of analytical grade. During the whole working process, water (>18.2 MΩ) treated by an ultra-pure water system (RFU 424TA, Advantech Aquarius, Irvine, CA, USA) was used. In addition, a water bath incubator (BT100, Yamato Kagaku Co., Ltd., Tokyo, Japan), a vacuum drying oven (DP33, Yamato Kagaku Co., Ltd., Tokyo, Japan), and a pH meter (HORIBA F-72, Tokyo, Japan) were used in this work. 

Furthermore, a scanning electron microscopy and energy dispersive spectrometry (SEM-EDS, JEOL, Akishima, Tokyo, Japan: JCM-6000 with JED-2300), ion sputtering (JFC-1100E, JEOL, Akishima, Tokyo, Japan), Fourier transform infrared spectroscopy (FT-IR4200, Jasco, Hachioji, Tokyo, Japan) nuclear magnetic resonance (NMR, AVANCE III HD400, Bruker, Billerica, MA, USA), Zeta-potential measurement with electrophoretic light scattering (ELS, Otsuka, Tokyo, Japan: ELSZ-2000ZS), and differential refractometry (RID-6A, SHIMADZU, Kyoto, Japan) were employed in this work. 

### 2.1. Synthesis of the Adsorbent

First, epichlorohydrin was grafted onto tertiary amines of dodecyl dimethyl to synthesize quaternary ammonium salts [23]. Concentrated hydrochloric acid was diluted to 4 M with ultra-pure water, and then dodecyl dimethyl tertiary amine was slowly dropped from a drip funnel in an ice bath until the ratio of hydrochloric acid to tertiary amine was 1:1, and the mixture was left to react for 1 h. Secondly, epichlorohydrin was slowly dropped into a flask at 35 °C for 30 min until the ratio of epichlorohydrin to tertiary amine was 1.5:1, and the mixture was left to react for 8 h. The reaction principle, based on Bao et al. [22], is shown in Figure 1. Next, ultra-pure water was added to remove the top layer of oil. The remaining liquid was extracted by ethyl acetate and chloroform successively. After extraction, the liquid was distilled at 100 °C, and the remaining liquid was considered to be a quaternary ammonium salt solution when the volume of the solution did not decrease.

The adsorbent was synthesized with quaternary ammonium cations and cellulose. At a ratio of 1:60, cotton balls were put into a bath of ultra-pure water and the quaternary ammonium salt solution was added at a concentration of 40 g·L^−1^. By heating the mixture to 75 °C, the cotton balls came in full contact with the quaternary ammonium salt. The mixture was then cooled to 50 °C, and sodium hydroxide was added at a concentration of 20 g·L^−1^ for 30 min [22,24]. The reaction principle, based on Bao et al. [22], is shown in Figure 2. Finally, the cotton balls were removed, rinsed thoroughly with pure water for 5 min, and dried in a drying oven at 60 °C to obtain cellulose modified with quaternary ammonium salts. 

### 2.2. Characterization of the Adsorbent

In this study, nuclear magnetic resonance (NMR) was first used to analyze the functional groups of quaternary ammonium salts dissolved in heavy water to determine whether the quaternary ammonium reaction was successful. After sputtering gold powder, the surface of the modified cellulose was observed by scanning electron microscopy-energy dispersive spectroscopy (SEM-EDS). The functional groups of cellulose before and after modification were analyzed by Fourier transform infrared spectroscopy (FT-IR) to determine whether the quaternary ammonium cation had modified the cellulose. Then, the zeta potential of cellulose before and after modification in different pH solutions was measured by electrophoretic light scattering (ELS). Finally, after the adsorption experiment, X-ray photoelectron spectroscopy (XPS) was used to analyze the elements of cellulose before and after the adsorption of SDS. 

### 2.3. Adsorption Experiments 

In order to study the optimal conditions of SDS adsorption, the effect of pH, adsorption time, and adsorption temperature on the SDS adsorption capacity of the modified cellulose was assessed. The modified cellulose was placed in a 200 mL conical flask containing 30 mL of an aqueous solution with a known amount of SDS, and the suspensions were placed in a constant temperature shaker (TAITEC Plus Shaker EP^−1^ with Thermo Minder SX-10R, Saitama, Japan) set at a prescribed temperature. Adsorption experiments were performed in the pH range of 3–11, with a contact time of 1 to 6 h, a temperature of 20–60 °C, and initial SDS concentrations of 100–600 mg·L^−1^. The pH of the test solution was adjusted by adding 0.1 mol·L^−1^ hydrochloric acid or sodium hydroxide. Finally, the adsorbed cellulose was taken out, and the residual concentration was measured by differential refractometry (RID) to calculate the adsorption capacity. The calculation method is shown in Equations (1) and (2).
(1)$$\;\;C_{t}=\frac{C_{0}}{n_{0}}\Delta n_{t}$$
(2)$$q_{t}=\frac{\left(C_{i}-C_{t}\right)\;V}{m}$$
where $q_{t}$ represents the adsorption capacity at equilibrium (mg·g^−1^); *C*_0_ is the concentration of SDS in the standard solution (mg·L^−1^); *C_i_* and *C_t_* are the initial and equilibrium SDS concentrations (mg·L^−1^) in the solution, respectively; *n_t_* and *n*_0_ are the tortuosity ratio of the adsorbed SDS and the SDS in the standard solution, respectively; *V* is the volume of the solution (L); and *m* is the mass of adsorbent (g) [25,26,27].

## 3. Results and Discussion

### 3.1. Characterization 

#### 3.1.1. NMR Analysis Results of the Quaternary Ammonium Salt

After the quaternary ammonium salt was dissolved in heavy water, nuclear magnetic resonance (NMR) was used to confirm the synthesis of the quaternary ammonium salt, and the results of this analysis are shown in Figure 3. According to an analysis of the chemical shift of different peaks, we can discover the kinds of functional groups hydrogen belongs to. The dodecyl group appeared at 0.5–2.0 ppm, the methyl group linked to nitrogen appeared at 3.1 ppm, the methylene group linked to nitrogen appeared at 3.3–3.4 ppm, the methylene group linked to chlorine appeared at 3.5–3.6 ppm and the methine group linked to the hydroxyl group appeared at 4.4 ppm [22]. Therefore, it can be seen from the measurements of the substance that the quaternary ammonium salts were indeed grafted with dodecyl dimethyl. 

#### 3.1.2. FT-IR Analysis Results of Cellulose

After making thin sheets of cotton floss with KBr, the structure of cotton floss was analyzed by Fourier transform infrared spectroscopy (FT-IR). The results are shown in Figure 4. It can be seen that the modified cellulose had a more obvious peak at 1637 cm^−1^ and 2900 cm^−1^ than pure cellulose. The peak at 1637 cm^−1^ was generated by the variable angular vibration of the quaternary ammonium group, and the peak at 2900 cm^−1^ was generated by the stretching vibration of methylene in the long-chain dodecyl group [28,29]. This confirmed that the quaternary ammonium group had modified the cellulose.

#### 3.1.3. SEM-EDS Analysis Results of Cellulose

After the cellulose was coated with gold powder by ion sputtering, the material was observed by scanning electron microscopy and energy dispersive spectrometry (SEM-EDS) at 15 kV voltage and under high vacuum conditions. The results of these samples [(a) pure cellulose, (b) modified cellulose, (c) adsorbed cellulose] are shown in Figure 5 (with the magnification of 2000). In the case of modified cellulose, the sample with the magnification of 220 (Figure 5b*) was also measured for the observation of the overall look of the sample. It was found that the quaternary ammonium salts adhered to the modified cellulose, making it coarser than pure cellulose. After adsorption, we can see that SDS is evenly wrapped around the fiber surface. The EDS results (Figure 6) showed the distribution of elements on the modified cellulose’s surface, which also indicated that the quaternary ammonium group was modified in the cellulose.

#### 3.1.4. XPS Analysis Results of Cellulose

The surface elemental information of cellulose was investigated by XPS and elemental analysis was carried out. The analysis results are shown in Figure 7 and Table 1. Although nitrogen was detected in the modified cellulose, it was low compared with the other elements (C and O). The result may be related to the fact that the proportion of nitrogen in the quaternary ammonium salt in cellulose is very low. The modification rate was estimated to be only 15.7%, which may be attributable to the large volume of dodecyl, and thus it was not easy to modify the cellulose. Moreover, the elemental composition of the adsorbed cellulose, including the detection of sulfur, suggested that the SDS can be truly absorbed by the modified cellulose.

### 3.2. Adsorption Experiments

#### 3.2.1. The Effect of pH

In order to investigate the effects of the solution’s pH on the uptake of SDS, adsorption experiments were conducted at different pH values (3, 5, 7, 9, and 11). As shown in Figure 8, the adsorption amount reached the maximum when the pH was around seven. At a lower pH value, the hydrogen ions compete with the quaternary ammonium cations, affecting the interaction between quaternary ammonium cations and SDS. On the other hand, when the pH was higher than seven, the hydroxide ions preferentially combined with quaternary ammonium cations, resulted in a deterioration in the adsorption capacity. Therefore, the pH was adjusted to seven for the removal of SDS by the modified cellulose for further experiments because the pH value of the solution was about nine before the adjustment.

#### 3.2.2. The Effect of Temperature

The effects of the reaction temperature on the SDS adsorption capacity of modified cellulose were investigated at different temperatures (20–70 °C). The results are shown in Figure 9. It can be seen that the adsorption capacity reached the maximum value when approaching 60 °C. Before 60 °C, with an increase in the reaction temperature, the adsorption amount generally increased, which indicated that the adsorption reaction was endothermic. After 60 °C, the adsorption amount decreased, which may be attributable to the high-temperature desorption of SDS that had been adsorbed through the intermolecular van der Waals force. Therefore, 60 °C was chosen for further experimental work.

#### 3.2.3. The Effect of Adsorption Time

The effect of contact time on the SDS adsorption capacity of modified cellulose was investigated. The results are shown in Figure 10. It can be seen that there was a significant increase from the beginning of adsorption up to 2 h, after that, there was no appreciable change. Therefore, the adsorption time was set at 3 h thereafter.

### 3.3. Adsorption Kinetics Study

The adsorption kinetics model can be used to determine the mechanism of the adsorption process and provide effective data support for a feasibility study of the process. In this experiment, the SDS adsorption capacity was investigated at 1, 5, 10, 20, 40, and 60 min under the optimal adsorption conditions. The experimental data were fitted into the pseudo-first- and pseudo-second-order reaction equations to study the mechanism of the adsorption process.

The pseudo-first-order rate equation is expressed as follows:(3)$$ln\left(q_{e}-q_{t}\right)=ln\left(q_{e}\right)-k_{1}t$$
where *q_e_* and *q_t_* are the adsorption capacity (mg·g^−1^) of SDS at equilibrium and at time *t*, respectively, and *k*_1_ is the rate constant (h^−1^) of pseudo-first-order adsorption.

The pseudo-second-order rate equation is expressed as follows:(4)$$\frac{t}{Q_{t}}=\frac{1}{k_{2}q_{e}^{2}}+\frac{1}{q_{e}}t$$
where *q_e_* and *q_t_* are the adsorption capacity (mg·g^−1^) of SDS at equilibrium and at time *t*, respectively, and *k*_2_ is the rate constant (g·mg^−1^·h^−1^) of pseudo-second-order adsorption [30].

In order to examine the consistency between the kinetics model and the experimental results, pseudo-first-order or pseudo-second-order models were applied to the linear graphs of *ln(q_e_−q_t_)* vs. *t* or those of *t*/*q_e_* vs. *t* for the SDS adsorption by modified cellulose, as shown in Figure 11 and Figure 12. Table 2 shows the linear constants (*R*^2^) and other parameters of the two kinetic models for the SDS adsorption by modified cellulose calculated according to Figure 11 and Figure 12. Judging the two values of *R^2^* shows that the adsorption process is highly correlated with both models. However, the maximum adsorption capacity calculated from the pseudo-second-order model was closer to that of the experimental data, so the adsorption of SDS is more consistent with the pseudo-second-order equation. Therefore, it is suggested that the adsorption reaction of SDS in this work was dominated by chemical adsorption as well as the van der Waals force on physical adsorption. The maximum adsorption capacity was estimated to be 33.6 mg·g^−1^.

### 3.4. Adsorption Isotherm Study

In the process of adsorption, the study of adsorption isotherms is significant and crucial for predicting the adsorption behavior of pollutants onto the adsorbent’s surface. In this experiment, typical adsorption isotherms, such as the Langmuir and Freundlich isotherms, were used to evaluate the experimental data of SDS adsorption. In this experiment, the adsorption capacity was explored with varying initial concentrations of 100, 200, 300, 400, 500, and 600 ppm. 

The Langmuir adsorption isothermal model assumes that the surface of the adsorbent is uniform and that the adsorbate is adsorbed on the surface of the adsorbent in the form of a single molecular layer. The Langmuir adsorption isotherm is given by Equation (5):(5)$$\frac{C_{e}}{q_{e}}=\frac{C_{e}}{q_{max}}+\frac{1}{K_{L}q_{max}}$$
where *C_e_* and *q_e_* are the SDS concentration (mg·L^−1^) and adsorption capacity (mg·g^−1^) when the adsorption reaches equilibrium, *q_max_* is the maximal adsorption capacity of the adsorbent (mg⋅g^−1^) and *K_L_* is the adsorption constant of the Langmuir isotherm (L·mg^−1^). The relationship between *C_e_*/*q_e_* and *C_e_* gives a straight line with a slope of 1/*q_max_* and an intercept of 1/(*K_L_*·*q_max_*). *K_L_* can be applied to the Gibbs adsorption free energy Δ*G_ads_* (J⋅mol^−1^) via Equation (6):*n*Δ*G_ads_* = −*RT lnK_L_*(6)
where *R* is the gas constant (8.314 J⋅K^−1^mol^−1^), *T* is the absolute temperature at equilibrium (K) and *K_L_* is the equilibrium constant at temperature *T*.

The equilibrium constant [31,32] *K_L_* can be calculated according to Equation (7):*K_L_* = *q_e_*/*C_e_*(7)
where *C_e_* and *q_e_* are the same as in Equation (4). 

The Langmuir constant (*K_L_*) can be used to determine the suitability of the adsorbent for the adsorbate by using the Hall separation coefficient (*R_L_*), a dimensionless parameter which is given by Equation (8):(8)$$R_{L}=(\frac{1}{1+K_{L}C_{0}})$$
where *C*_0_ (mg⋅L^−1^) is the initial concentration. When 0 < *R_L_* < 1, this means that the adsorbent is suitable for the adsorbate [33,34].

The Freundlich adsorption isothermal model is a multilayer adsorption process that does not consider adsorption saturation, which occurs on a multi-layered heterogeneous surface. The linear Freundlich isotherm model is represented by Equation (9):(9)$$ln\;q_{e}=ln\;K_{F}+\left(1/n\right)ln\;C_{e}$$
where *K_F_* is the adsorption capacity((mg·g^−1^) and 1/*n* is the adsorption strength. The relationship between *lnq_e_* and *lnC_e_* can determine the values of 1/*n* and *K_F_*. The value of 1/*n* can be used to judge the difficulty of the adsorption process as follows: irreversible adsorption, 1/n=0; favorable adsorption, 0<1/n<1; unfavorable adsorption, 1/n>1 [33].

The results of the adsorption data investigated by the Langmuir and Freundlich equations are shown in Figure 13 and Figure 14, respectively. The value of *K*_L_ can be calculated from the intercept of the Langmuir isotherm in Equation (4). *K_L_* represents the affinity strength of the adsorbent to the target substance [35].

Table 3 shows the linear constants (*R*^2^) and other parameters of the two isotherm models for the modified cellulose adsorption of SDS calculated according to Figure 13 and Figure 14. By comparing the two values of *R^2^*, we can see that the adsorption isotherms can generally be described more satisfactorily by the Langmuir isotherm. In addition, the value of *R_L_* is between 0 and 1 when the initial concentration is 100 mg·L^−1^. This result suggests that the adsorption of SDS on the modified cellulose mainly occurred by a monolayer reaction with the strong adsorption force. The maximum adsorption capacity was estimated to be 22.8 mg·g^−1^.

### 3.5. Adsorption Thermodynamics 

In the adsorption process, the direction of heat flow and the reaction in the adsorption process can be guided by a study of the adsorption thermodynamics, which provides theoretical support for improving the reaction’s efficiency. In this experiment, under the optimal adsorption conditions, the SDS adsorption capacity was investigated at adsorption temperatures of 293, 313, 333, and 343 K. The Gibbs free energy, enthalpy change, and entropy change in the SDS adsorption reaction were calculated by the Van’t Hoff equation. The Van’t Hoff equation is as follows (Equation (10)):(10)$$\;\Delta G^{0}=\Delta H^{0}-T\Delta S^{0}$$

The simultaneous equations (Equations (5) and (6)) lead to the following formula (11):(11)$$ln\frac{q_{e}}{C_{e}}=\frac{\Delta S^{0}}{R}-\frac{\Delta H^{0}}{RT}$$
where *q_e_*, *C_e_*, *R,* and *T* are the same as in Equations (5) and (6), whereas ∆*S*^0^ (J·mol^−1^), ∆*H*^0^ (kJ·mol^−1^), and ∆*G*^0^ (kJ·mol^−1^) represent the changes in the entropy, enthalpy, and Gibbs free energy during the reaction, respectively. The relationship between *ln(q_e_*/*C_e_)* and 1/*T* shown in Figure 15 gives a straight line with a slope of ∆*H*^0^/*R* and an intercept of ∆*S*^0^/*R* [30]. The thermodynamic parameters calculated via the Van’t Hoff equation are shown in Table 4. It can be seen that the change in the Gibbs free energy has a negative value, but it tends to decrease with an increase in the temperature, indicating that the adsorption reaction of SDS is spontaneous. From the positive value of the change in enthalpy, it can be concluded that the adsorption reaction of SDS is endothermic. The change in entropy has a positive value, indicating that the degree of randomness of the interface between the adsorbent and the solution increases as the reaction progresses.

### 3.6. The Zeta-Potentials of Modified Cellulose Samples 

Zeta potential vs. pH data of modified cellulose at different pH is shown in Figure 16. In our work, the concentration of NaCl was installed 0.01 mol/L based on Li’s work [36], and the solid–liquid ratio was 1:2000 (i.e., 25 mg for modified cotton). After the solution pH was adjusted to 1.8–6.8 using HCl and NaOH, the zeta potential of each sample was measured by electrophoretic light scattering [37].

From Figure 16, firstly, it can be seen that the zeta potential of the cotton, which was modified by quaternary ammonium cations is positive in the interval of pH 1.8–6.8, while the pure cotton shows negative values higher than pH 2.5 [38]. This is the reason why the modified cotton has a stronger attraction to the anionic group compared to the pure cotton. After reaching a peak at pH 3, the zeta potential decreases with an increase in pH value. 

As shown in Figure 11 (in Section 3.2.1), the adsorption of SDS, which is largely influenced by the concentration of hydrogen ions, is maximum at pH 7. At a low value of pH, it is considered that too many hydrogen ions may compete with quaternary ammonium cations, leading to a decrease in the adsorption of SDS, although the zeta potential is high.

### 3.7. Comparison with Other Adsorbents

A comparison of the maximum SDS adsorption capacity in the present study against those of other adsorbents in the literature is presented in Table 5. As can be seen from Table 5, the SDS adsorption capacity of the modified cellulose in this work (32.5 mg·g^−1^) was not necessarily high compared with other adsorbents. However, it could be regarded as a potential adsorbent for removing SDS from wastewater for practical use because the method of synthesizing the adsorbent is relatively simple, and cellulose as a raw material is readily available for the removal of SDS. 

Moreover, the modified cellulose can be expected to have better adsorption performance if the modification conditions are optimized to improve the modification rate, although the modification rate of quaternary ammonium cations was only 15.7% in this work.

## 4. Conclusions

In this work, the synthesis of long-chain dodecyl in a quaternary ammonium salt could be confirmed at the laboratory level through NMR analysis. The results of SEM-EDS and FT-IR showed that the quaternary ammonium cation was successfully modified in cellulose. The optimal conditions of SDS adsorption by quaternary ammonium cation-modified cellulose were pH 7, an adsorption time of 3 h, and a reaction temperature of 60 °C. The analysis of the adsorption kinetics showed that the adsorption process of SDS was more consistent with the pseudo-second-order model, indicating that the adsorption was mainly dominated by chemical adsorption. The adsorption process of SDS conformed more to the Langmuir isotherm, suggesting that SDS was mainly adsorbed on the surface of the cellulose in the form of a monolayer. According to the analysis of the adsorption thermodynamics, the adsorption reaction of SDS is a spontaneous and endothermic reaction. From the results of zeta potential, it is clear that the modification of quaternary ammonium salt makes the cellulose surface have a more positive charge, leading to a stronger attraction to SDS. Finally, it can be concluded that the modified cellulose with quaternary ammonium cations synthesized in this study can be an effective adsorbent for SDS from an aqueous solution.

## Figures and Tables

**Figure 1 polymers-14-01473-f001:**
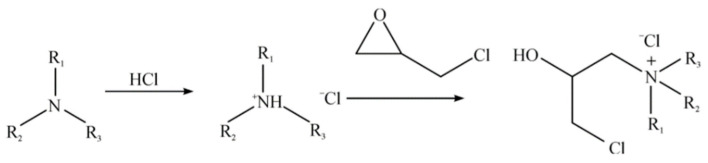
The reaction process for synthesizing the quaternary ammonium salt.

**Figure 2 polymers-14-01473-f002:**
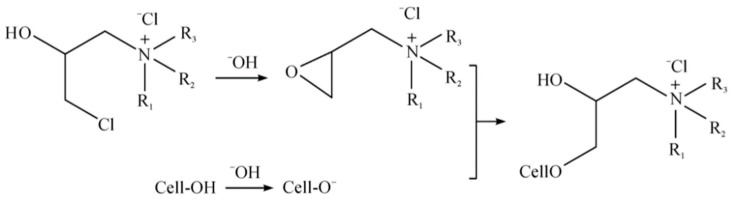
The reaction process of modifying the cellulose.

**Figure 3 polymers-14-01473-f003:**
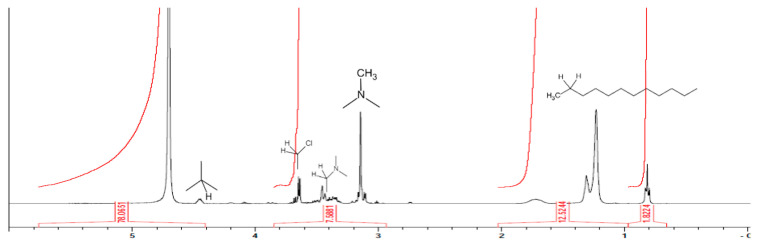
NMR spectra of (3-chloro-2-hydroxypropyl) chloride.

**Figure 4 polymers-14-01473-f004:**
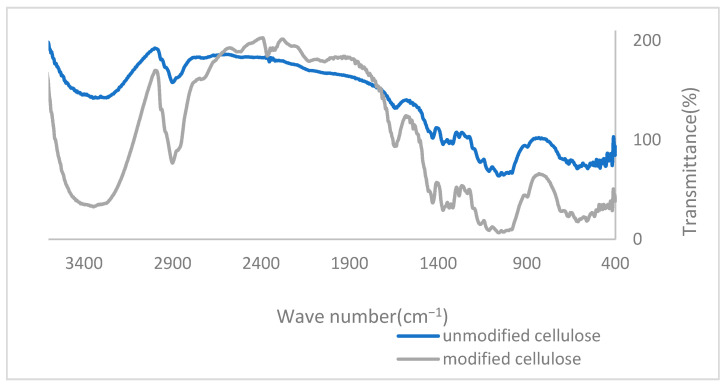
FT-IR spectra of cellulose and modified cellulose.

**Figure 5 polymers-14-01473-f005:**
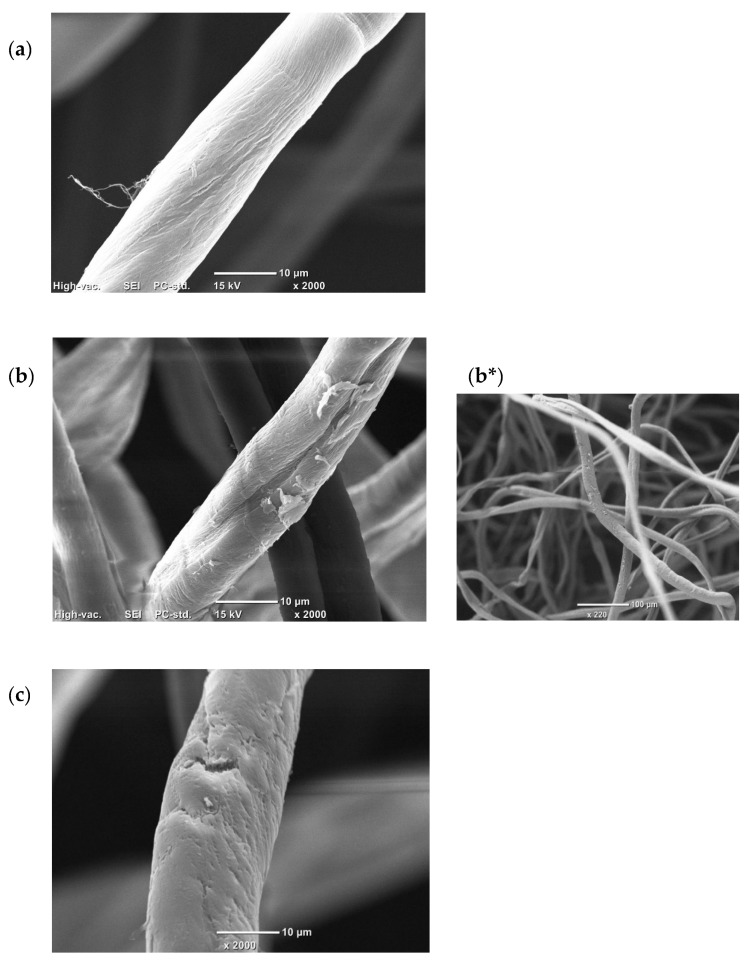
SEM image of (**a**) the pure cellulose (×2000), (**b**) the modified cellulose (×2000) [(**b***) (×220)], (**c**) the adsorbed cellulose (×2000).

**Figure 6 polymers-14-01473-f006:**
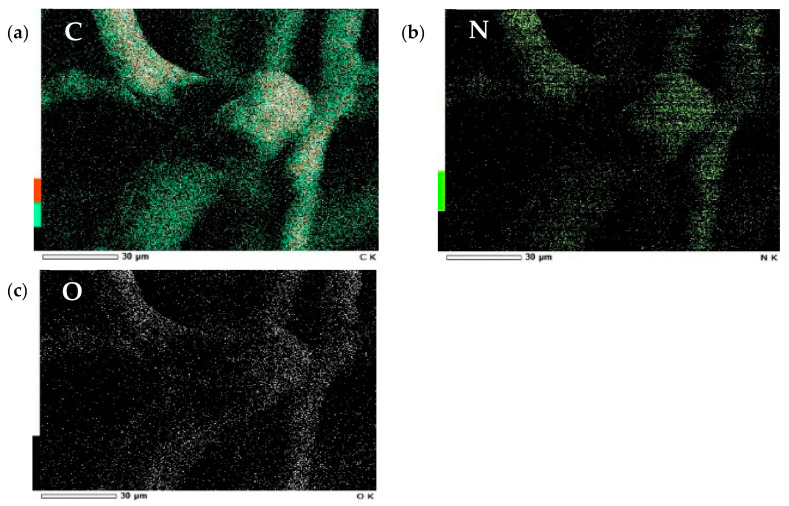
Element mapping images of modified cellulose [(**a**): C, (**b**): N, (**c**): O].

**Figure 7 polymers-14-01473-f007:**
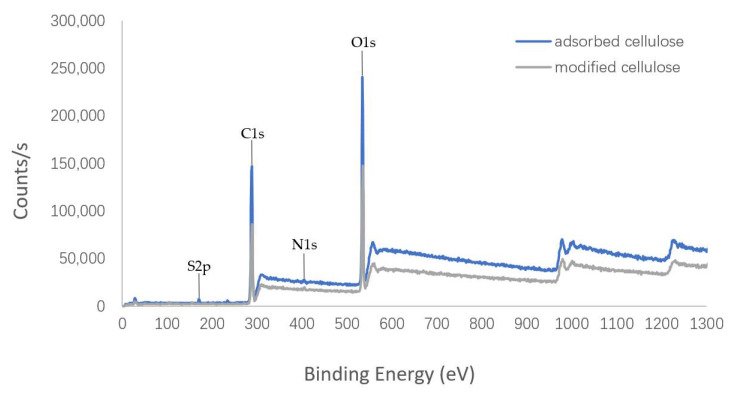
XPS image of cellulose.

**Figure 8 polymers-14-01473-f008:**
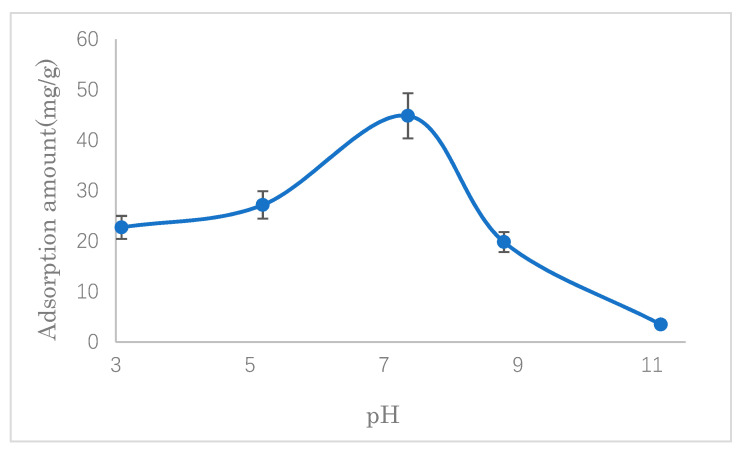
Effect of pH on SDS adsorption.

**Figure 9 polymers-14-01473-f009:**
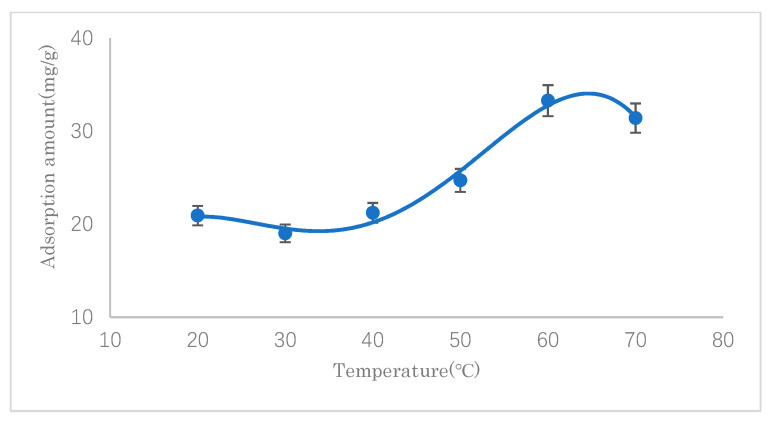
Effect of temperature on SDS adsorption.

**Figure 10 polymers-14-01473-f010:**
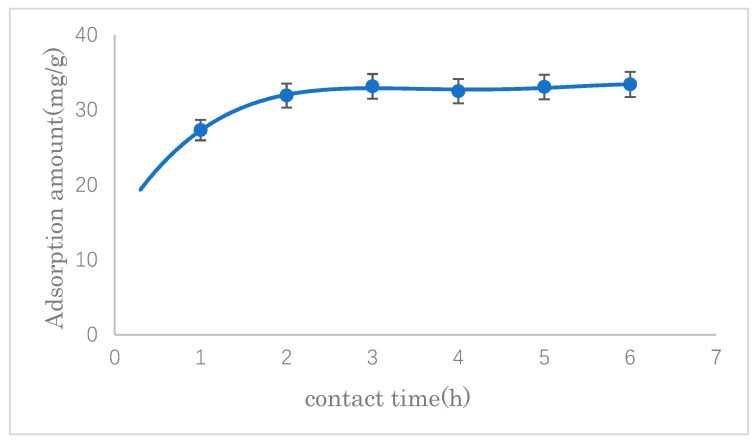
Effect of contact time on SDS adsorption.

**Figure 11 polymers-14-01473-f011:**
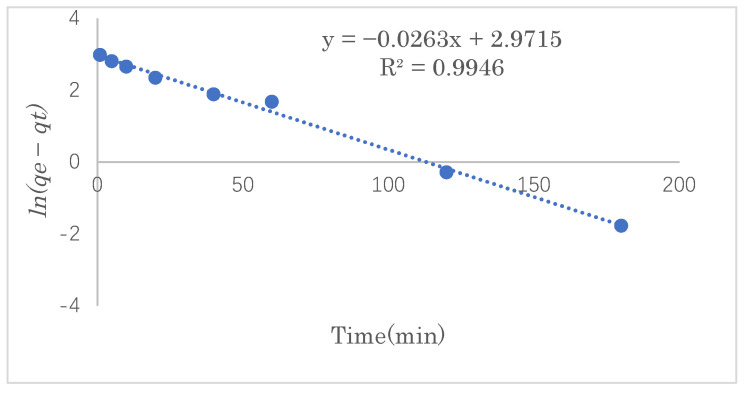
Pseudo-first-order plot of SDS adsorption by modified cellulose.

**Figure 12 polymers-14-01473-f012:**
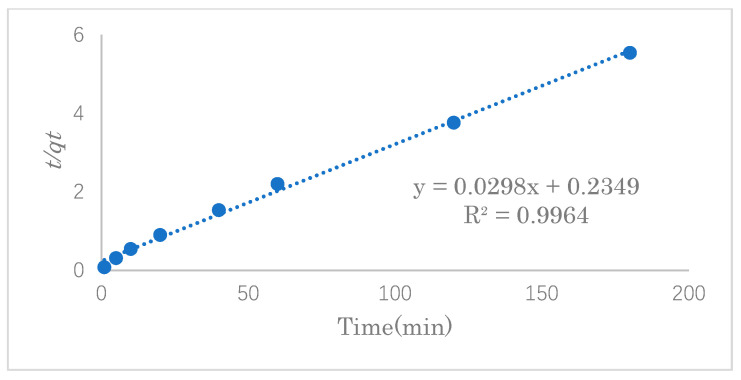
Pseudo-second-order plot of SDS adsorption by modified cellulose.

**Figure 13 polymers-14-01473-f013:**
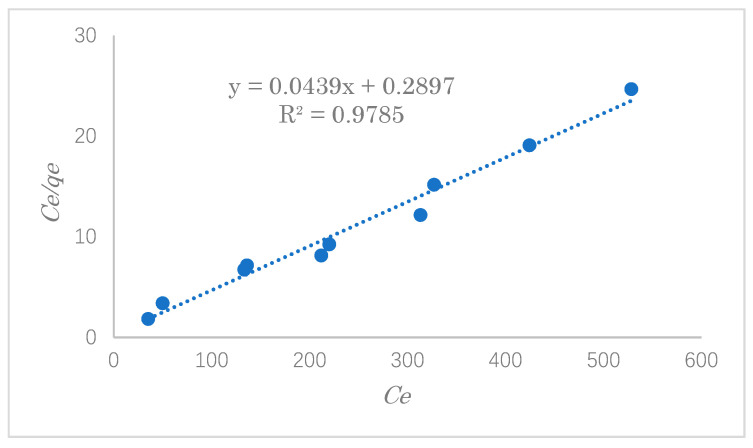
The Langmuir isotherm for the adsorption of SDS by modified cellulose.

**Figure 14 polymers-14-01473-f014:**
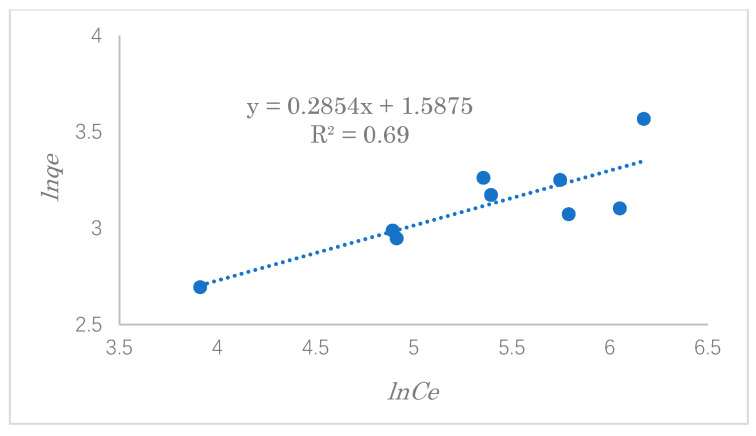
The Freundlich isotherm for the adsorption of SDS by modified cellulose.

**Figure 15 polymers-14-01473-f015:**
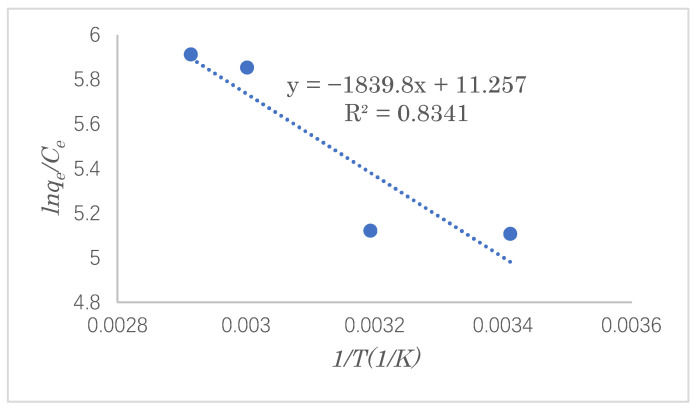
Plot of $ln\frac{q_{e}}{C_{e}}$ and 1/*T* for estimation of the thermodynamic parameters obtained for the adsorption of SDS by modified cellulose.

**Figure 16 polymers-14-01473-f016:**
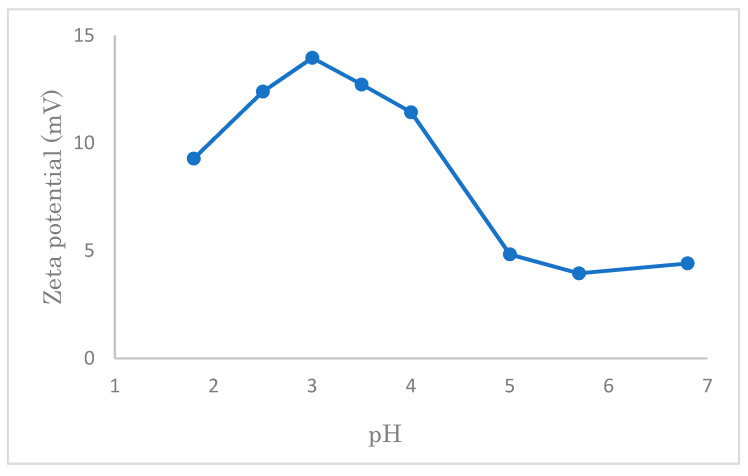
Zeta potential of modified cellulose (in 0.01 mol/L NaCl) at different pH.

**Table 1 polymers-14-01473-t001:** The proportional of element in the cellulose.

Cellulose	C%	O%	N%	S%
Modified	63.9	33.8	0.93	-
Adsorbed	66.1	32.2	0.62	1.04

**Table 2 polymers-14-01473-t002:** Isotherm parameters of SDS adsorption by the modified cellulose.

		Pseudo-First-Order Model			Pseudo-Second-Order Model		
Target	*q_eep_* (mg/g)	*q_e_* (mg/g)	*k*_1_ (min^−1^)	*R* ^2^	*q_e_* (mg/g)	*k*_2_ (g/mg min^−1^)	*R* ^2^
SDS	32.5	19.5	0.0263	0.995	33.6	0.00378	0.996

**Table 3 polymers-14-01473-t003:** Isotherm parameters for SDS adsorption by modified cellulose.

Target	*T* (°C)	Langmuir Isotherm				Freundlich Isotherm		
		*q_max_* (mg/g)	*R_L_*	*K_L_*	*R^2^*	*K_F_* (mg/g)	1/*n*	*R^2^*
SDS	60	22.8	0.0619	0.354	0.979	4.89	0.0285	0.690

**Table 4 polymers-14-01473-t004:** Thermodynamic parameters for the adsorption of SDS by modified cellulose.

*T* (K)	Δ*H* (kJ/mol)	Δ*S* (J/mol)	Δ*G* (kJ/mol)
293	15.29	93.59	−12.14
313	-	-	−14.01
333	-	-	−15.88
343	-	-	−16.82

**Table 5 polymers-14-01473-t005:** Comparison of the adsorption capacity of different adsorbents.

Adsorbent	*q_max_* (mg·g^−1^)	Reference
Activated coconut shell	111	[11]
Silica	0.202	[36]
Chitosan hydrogel beads	76.9	[39]
Aquaguard waste activated carbon	61.5	[40]
Granular activated charcoal	3.75	[41]
Pine cone bio mass	95.8	[42]
Kaolinite	1.50	[43]
Modified cellulose	32.5	This study

## Data Availability

Not applicable.

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
