# Peer review of "Adsorption of an Anionic Surfactant (Sodium Dodecyl Sulfate) from an Aqueous Solution by Modified Cellulose with Quaternary Ammonium"

_polymers, 2022, doi:10.3390/polym14071473_

Round 1

Reviewer 1 Report

The manuscript titled "Adsorption of an anionic surfactant (sodium dodecyl sulfate) from an aqueous solution by modified cellulose with quaternary ammonium" is generally well written although the novelty is not very clear.

  1. In the introduction provide more information about the novelty of the the process. there have been several research on this subject.
  2. check the labels in Figure 8
  3. Were the experiments done in duplicate? no error bars in Figures 8 to 11.
  4. A significant amount of the references are very old. improve on this.

Author Response

Thank you for your comment and valuable suggestion.

(1) In the introduction provide more information about the novelty of the the process. there have been several research on this subject.

Response: Thank you for your comment and valuable suggestion. The novelty of the process in our present work is using “dodecyl dimethyl amine” for synthesizing modified cellulose with quaternary ammonium. Tri-methyl amine have been used as the raw material for the quaternary ammonium salt in most of the studies as reference [33] in revised manuscript ([21] in original manuscript). It is expected that dodecyl can interact with hydrophobic base of SDS, and then lead to improve the adsorption efficiency. However, few studies have been conducted using dodecyl-modified cellulose for SDS adsorption. Based on your suggestions, we have added the description for selecting “dodecyl dimethyl amine” in Introduction section in our revised manuscript.

(2) Check the labels in Figure 8.

Response: Thank you for your comment. We have checked Figure 8 (i.e., Figure 6 in the revised manuscript) and have revised the labels.

(3) Were the experiments done in duplicate? no error bars in Figures 8 to 11.

Response: Thank you for your comment and valuable suggestion. The experiments are performed in duplicate. Based on your suggestion, we have added the error bars in Figures 8-10 (in the revised manuscript).

(4) A significant amount of the references are very old. improve on this.

Response: Thank you for your comment and valuable suggestion. Based on your suggestion, we have added recent 7 literatures as [17-19] (in addition to [12-15] regarding MOFs) in our revised manuscript.     

Reviewer 2 Report

In this manuscript, the authors investigated the SDS adsorption efficiency of modified cellulose in water. The manuscript is well written. I have some minor suggestions.

  1. MOFs as promising porous materials that can also be used as novel adsorbents and catalyst. Some references can be cited in the introduction section, for example Journal of Water Process Engineering 44 (2021): 102437 (SDS separation from aqueous solutions using ZIF-8/Carbon fiber); Chemistry–A European Journal26(61), 13788-13791 (Reaction of epichlorohydrin with CO2 catalyzed by amine functionalized MOF).
  2. Figure 3, I suggest the authors to draw the chemical structure and label the peaks in the NMR with the protons on the structure. In addition, the peak integration and peak position should be clearly presented in the figure. 
  3. Figure 4, For better comparison, I suggest the authors to extract substrate and normalize the peak. Also, shouldn't the maximum of transmittance be 100%? Why the y axis shows 500% transmittance?
  4. For SEM images, how the width of cellulose changes after modification?
  5. Figure 9, there also seems to be a small peak corresponds to S2p in Figure 9. I suggest the authors combine Figure 9 and Figure 10 in one figure for better comparison. 

Author Response

Thank you for your comment and valuable suggestion. 

(1) MOFs as promising porous materials that can also be used as novel adsorbents and catalyst. Some references can be cited in the introduction section, for example Journal of Water Process Engineering 44 (2021): 102437 (SDS separation from aqueous solutions using ZIF-8/Carbon fiber); Chemistry–A European Journal26(61), 13788-13791 (Reaction of epichlorohydrin with CO2 catalyzed by amine functionalized MOF).

Response: Thank you for your comment and valuable suggestion. Based on your suggestion, we have added some descriptions about MOFs in Introduction in our revised manuscript, and cited 4 articles including your suggested 2 articles ([12]- [15] in our revised manuscript).

(2) Figure 3, I suggest the authors to draw the chemical structure and label the peaks in the NMR with the protons on the structure. In addition, the peak integration and peak position should be clearly presented in the figure. 

Response: Thank you for your comment and valuable suggestion. Based on your suggestion, we have revised Figure 3.

(3) Figure 4, For better comparison, I suggest the authors to extract substrate and normalize the peak. Also, shouldn't the maximum of transmittance be 100%? Why the y axis shows 500% transmittance?

Response: Thank you for your comment and valuable suggestion. We have understood and agreed with your idea. Based on your suggestion, we have revised Figure 4 because the region after 3400cm-1 is meaningless part.

We considered that our cotton fiber sample film of uneven thickness may make the transmittance higher than that of the KBr thin film (used as a reference), leading to transmittance greater than 100%.

(4) For SEM images, how the width of cellulose changes after modification?

Response: Thank you for your comment and valuable suggestion. We have roughly estimated the pure cellulose for the width of 13.95 μm, modified cellulose for the width of 14.08 μm and adsorbed cellulose for the width of 19.14 μm at the same magnification from SEM images. However, it is difficult to compare the width among these samples exactly, and clearly changes was not observed at present.  

(5) Figure 9, there also seems to be a small peak corresponds to S2p in Figure 9. I suggest the authors combine Figure 9 and Figure 10 in one figure for better comparison. 

Response: Thank you for your comment and valuable suggestion. Based on your suggestion, we have combined the two figures in one figure (Figure 7 in our revised manuscript) for better comparison. Comparing the position of S2p in these figures, the peak only after adsorption has more clearly peak at about 170 eV.

Reviewer 3 Report

In this current manuscript, the authors aimed for removing sodium dodecyl sulfate (SDS) from an aqueous solution using the modified cellulose with quaternary ammonium cation. The manuscript can be accepted after checking the following comments.

Abstract has meaningless data such as the source of the utilized chemicals and the utilized tools for characterization. In addition, there is no data in the abstract for the modified cellulose. Overall, abstract should be completely rewritten.

Country source for the utilized cotton is missing.

The introduction part should be rewritten for achieving the full story and it should be enhanced with these relative references:

- Improvement of enzymatic properties and decolorization of azo dye: Immobilization of horseradish peroxidase on cationic maize starch - Synthesis, characterization and adsorption properties of microcrystalline cellulose based nanogel for dyes and heavy metals removal - Remediation of Cd (II) and reactive red 195 dye in wastewater by nanosized gels of grafted carboxymethyl cellulose

SEM images (Figure 5, 6, and 7) must be taken at the same magnification.

Weigh percent for the presented elements; C, N, and O appeared in EDX images should be added.

XRD of the unmodified and modified samples should be added.

SEM for the adsorbent cellulose after treatment must be added to investigate the morphological surface structure after SDS removal.

Author Response

Thank you for your comment and valuable suggestion. 

(1) Abstract has meaningless data such as the source of the utilized chemicals and the utilized tools for characterization. In addition, there is no data in the abstract for the modified cellulose. Overall, abstract should be completely rewritten.

Response: Thank you for your comment and valuable suggestion. Based on your suggestion, we have added the main results regarding the modified cellulose in Abstract in our revised manuscript (The revised parts have been highlighted in blue letter in the revised manuscript).

(2) Country source for the utilized cotton is missing.

Response: Thank you for your comment and valuable suggestion. We have added the country source of cotton in Materials and Methods in our revised manuscript.

(3) The introduction part should be rewritten for achieving the full story and it should be enhanced with these relative references:

 -Improvement of enzymatic properties and decolorization of azo dye: Immobilization of horseradish peroxidase on cationic maize starch - Synthesis, characterization and adsorption properties of microcrystalline cellulose based nanogel for dyes and heavy metals removal - Remediation of Cd (II) and reactive red 195 dye in wastewater by nanosized gels of grafted carboxymethyl cellulose

Response: Thank you for your comment and valuable suggestion. Based on your suggestions, we have cited these literatures [17-19] (in our revised manuscript) to complete the introduction part of this paper.

(4) SEM images (Figure 5, 6, and 7) must be taken at the same magnification. Response: Thank you for your comment and valuable suggestion. Based on your suggestion, we have re-measured the pure cellulose, the modified cellulose and the SDS-adsorbed cellulose at the same magnification (x 2000), and have been shown in Figure 5 in the revised manuscript.

(5) Weigh percent for the presented elements; C, N, and O appeared in EDX images should be added.

Response: Thank you for your comment and valuable suggestion. We have tried to conduct the elemental analyses by EDX. However, exact data was not obtained because the peaks of S element overlapped with the gold elements plated on the surface of the cellulose. Then we have measured weigh percent for the presented elements (C, N, O and S) by XPS and shown in Table 1 in our manuscript.

(6) XRD of the unmodified and modified samples should be added.

Response: Thank you for your comment and valuable suggestion. We also consider that XRD analysis of these samples are important. However, this manuscript put most important point for the adsorption of an anionic surfactant (sodium dodecyl sulfate) by modified cellulose, and the information for crystallinity of cellulose was not particularly needed because we did not regard them to be the samples with clear crystal characteristics.

Then we did not carry out XRD analysis in this work, and detail investigation of the crystallinity of cellulose will be presented elsewhere in the future paper.

(6) SEM for the adsorbent cellulose after treatment must be added to investigate the morphological surface structure after SDS removal.

Response: Thank you for your comment and valuable suggestion. Based on your suggestions, we have newly added SEM images of the adsorbent cellulose (i.e, after adsorption of SDS) as Figure 5(c) in the revised manuscript.
